# Non-DRE Voided Urine Test to Diagnose Prostate Cancer: Updated Results

**DOI:** 10.3390/diagnostics15050607

**Published:** 2025-03-03

**Authors:** Patrick T. Gomella, Joon Yau Leong, Leonard G. Gomella, Vivek S. Tomar, Hector Teran, Edouard J. Trabulsi, Madhukar L. Thakur

**Affiliations:** 1Department of Urology, Thomas Jefferson University, Philadelphia, PA 19107, USA; jxl297@jefferson.edu (J.Y.L.); leonard.gomella@jefferson.edu (L.G.G.); 2Department of Radiology, Thomas Jefferson University, Philadelphia, PA 19107, USA; vxt148@jefferson.edu (V.S.T.); hector.teran@jefferson.edu (H.T.); 3Jefferson Einstein Medical Center, Thomas Jefferson University, Philadelphia, PA 19141, USA; edouard.trabulsi@jefferson.edu; 4Department of Radiology, Radiation Oncology and Urology, Thomas Jefferson University, Philadelphia, PA 19107, USA; madhukar.thakur@jefferson.edu

**Keywords:** prostate cancer, genomics, VPAC receptors, urine

## Abstract

**Background:** The standard diagnostic approach for prostate cancer (PCa) diagnosis consists of serum prostate-specific antigen (PSA) testing, digital rectal examination (DRE) and image-guided targeted biopsies. Given the invasive nature, potential adverse events and costs associated with these techniques, alternative approaches have been investigated, specifically with serum and urine assays. The work presented here is intended to further validate a novel noninvasive optical technique for PCa detection, targeting the VPAC genomic receptors that are overexpressed on prostate cancer’s malignant cells (MC), in non-DRE voided urine. **Methods:** Patients (*N* = 62) who had image-guided biopsy and histologically confirmed localized PCa, and who were scheduled for radical prostatectomy, provided a non-DRE voided urine sample prior to surgery. Urine was cytocentrifuged and cells fixed on a glass slide, incubated with 0.5 μg TP4303 (a receptor-specific fluorophore developed in our laboratory with high affinity for VPAC), excess washed and treated with 4,6-diamidodino-2-phenylindole (DAPI) for nuclear staining. The field of cells on each slide was analyzed using a Zeiss AX10 Observer microscope (20×). The total number of cells and MC were then counted, and the florescent intensity around each MC was measured using Zeiss software. Additionally, non-DRE voided urine samples collected from clinically determined BPH patients (*N* = 97), were also analyzed similarly. **Results:** Urine samples from 62 patients were processed and analyzed. Mean PSA levels by Gleason grade (GG) group were 6.5 ± 4.1 ng/mL for GG1 (*N* = 10), 7.2 ± 3.8 for GG2 (*N* = 31), 13.2 ± 14.6 for GG3 (*N* = 13), 6.2 ± 2.2 for GG4 (*N* = 2) and 50.2 ± 104.9 for GG5 (*N* = 6). Like the PSA, % MC shed (66.7 ± 27.7) in voided urine and the fluorescent intensity (35.8 ± 5.7) were highest in patients with GG5 prostate cancer. All PCa patients in GG1 to GG5 shed MC in voided urine with increasing % of MC and increasing fluorescence intensity which correlated with the increasing GG for PCa. For BPH, the specificity for the assay was 89.6% (95% CI:81.9–94.9%), PPV was 0.0% and NPV was 100% (95.9% CI, 95.9–100%). **Conclusions:** These data indicate the following: (i) PCa MC shed in non-DRE voided urine can be detected by targeting VPAC receptors, (ii) MC are shed in non-DRE voided urine with increasing quantity, corresponding to the severity of the disease, and (iii) this non-DRE voided urine optical assay provides a simple, noninvasive, and reliable method for the preliminary detection of PCa with potentially a lower cost than the currently available pre-biopsy detection technologies.

## 1. Introduction

Prostate cancer (PCa) is the most common non-cutaneous malignancy affecting men in the United States and is a leading cause of cancer-related mortality globally. In 2024, it was estimated that there will be 299,010 new diagnoses and 35,250 deaths due to PCa [1]. Evaluation of the serum prostate-specific antigen (PSA) test remains the most commonly used biomarker for the detection and follow-up of PCa. The test is sensitive, relatively inexpensive and is standardized. Since the Food and Drug Administration’s approval of PSA for screening in 1994, PCa disease-specific mortality has reduced [2,3,4]. However, the PSA test suffers from poor specificity. Therefore, while the mortality has improved to some degree, the use of PSA has led to high rates of overdiagnosis and overtreatment of indolent cancers. Furthermore, due to its low specificity, PSA measurements can lead to 70–80% unnecessary biopsies in the general male population [2]. While the advent of pre-biopsy multiparametric magnetic resonance imaging (mpMRI) and targeted biopsy have improved the selection and diagnostic yield of prostate biopsy [5], currently, the only way to diagnose prostate cancer is with invasive biopsy for tissue acquisition to facilitate histologic evaluation, considered as the gold standard. A noninvasive test for PCa diagnosis would be most welcome to patients and clinicians. Tests such as PCA3, 4K score, ExoDx and My Prostate Score have been developed to assist in prebiopsy risk stratification and in the management of patients with persistently elevated PSA after negative biopsy. Several of these tests require digital rectal examination (DRE) prior to urine collection. While these tests have shown promising results, their use is not yet widespread due to low positive predictive values, high cost and varying insurance coverages. Biopsy therefore still remains paramount [6]. As a result, the need exists for a more sensitive, specific, and noninvasive voided urine-based test that might be suitable for both clinical and widespread screening purposes, as well as for potentially predicting disease aggressiveness prior to biopsy.

VPAC (VPAC1 and VPAC2) are the members of the class B receptors belonging to the superfamily of GPCR [7]. These receptors are expressed in high density (10^4^–10^5^/MC) at the onset of oncogenesis but prior to the alteration of cell morphology, required for determination of PCa by histology [7,8,9,10,11,12]. On stroma, normal cells, inflammatory cells and benign masses, VPAC is only minimally present (~10/cell) [13,14,15]. We have designed, synthesized and validated a 28-amino-acid peptide, labeled it with a near-infrared (NIR) fluorophore and named it TP4303, which avidly (kd 3.1 × 10^−8^ M) binds to VPAC receptors [13,14,15]. Cells are collected on a glass slide from voided urine of a PCa patient, stained with TP4303 and observed with a fluorescence microscope, allowing us to detect PCa with high sensitivity [16]. The assay detected VPAC-positive cells in 98.6% of the patients with a PCa diagnosis, and none (0%) of the males with a benign prostatic hyperplasia (BPH) diagnosis alone. A follow-up evaluation showed 100% histopathologic correlation with no false positive or false negative results when shed cells were evaluated by targeting the VPAC receptor.

In addition to PCa, another ubiquitous condition in men between the ages of 50 and 60 is benign prostatic hyperplasia (BPH). The risk of having BPH in this age group is 50%, which increases to 70% in men 60–70 years of age The risk of having PCa in this age group is 17%. PCa and BPH co-exist in 20% of men older than 50, confusing early diagnosis and ultimately inducing considerable morbidity and even mortality in men, worldwide. As life expectancy continues to increase, this conundrum will only exacerbate progressively. While BPH and PCa are two distinctly different diseases, they share common pathways for certain conditions such as androgen dependance and inflammatory components. Much is known about their biochemical and genetic pathways that permit different approaches for management of PCa and BPH. However, their accurate and early diagnosis remains imperative. DRE and PSA do not distinguish PCa and BPH accurately and reliably.

In this prospective study, we aim to further validate our voided urine optical assay targeting the VPAC receptor to detect PCa, with additional evaluation based on the whole mount pathologic diagnosis after radical prostatectomy and to determine if the assay can distinguish PCa from BPH.

## 2. Patient and Methods

Study Population: This prospective study was conducted under an institutional review board (IRB)-approved protocol. Consenting patients with histologically confirmed any-grade PCa, scheduled for radical prostatectomy, were enrolled. Exclusion criteria included urinary tract infection, hematuria, a history of urothelial carcinoma, or prior pelvic radiotherapy. All patients provided a non-DRE voided urine sample (15–50 mL), which was processed within four hours at room temperature or stored at −10 °C for up to 72 h in case of delays.

Patients (*N* = 62, 63 to 73 years of age) who had image-guided biopsy and histologically confirmed localized PCa, who were also scheduled for radical prostatectomy, provided a non-DRE voided urine sample prior to surgery. In addition, non-DRE voided urine samples were also obtained from an additional 97 men, 50–70 years of age, who were clinically diagnosed with BPH and had PSA < 1.5 ng/mL (0.7 + 0.4 ng/mL). Their urine was processed for the assay in a similar manner. The results were evaluated statistically.

Sample Preparation and Imaging: As described previously [16] urine samples were centrifuged at 2000× *g* for 10 min, and all but approximately 250 μL of supernatant was discarded. Cells were resuspended, cytocentrifuged and followed by fixation with 97% ethanol. Slides were then incubated for 15 min with 0.5 μg TP4303, in 150 μL phosphate buffer. Excess TP4303 was then washed with deionized water, slides air dried and 50 μL of 4,6-Dimidino-2-phenylindole, Dihydrochloride (DAPI, Fisher Scientific, Norristown, PA, USA), a nuclear stain, were added onto the cells. After 20 min of incubation, a cover slip was placed, and the cells were examined using a fluorescence microscopy (Zeiss AX10 Observer, Zeiss, Oberkochen, Germany) at 20× magnification. VPAC-positive malignant PCa cells (MCs) were indicated by orange fluorescence around the nucleus. VPAC-negative cells, including epithelial cells, would only be stained by the DAPI, appearing dark blue (Figure 1). The total number of cells and the number of MCs were counted, and their fluorescence intensity was quantified using Zeiss software(Zen 3.2 Pro) (Figure 2). MC counts and fluorescence intensities were then compared with PSA levels and grade groups (GGs) as determined by postsurgical histopathology.

## 3. Results

Postsurgical histopathology confirmed that all patients in the cancer group who provided urine were positive for PCa. The severity of their disease (GG) was determined independently by genitourinary pathologists. Our results are summarized in detail in Table 1. All PCa patients in GG1 to GG5 shed MC in voided urine. PSA values were 6.5 ± 4.1 ng/mL for GG1 (*N* = 10), 7.2 ± 3.8 for GG2 (*N* = 31), 13.2 ± 14.6 for GG3 (*N* = 13), 6.2 ± 2.2 for GG4 (*N* = 2) and 50.2 ± 104.9 for GG5 (*N* = 6). PSA values were highest (50.2 ± 104.9) for GG5 (*N* = 6). Similar to the PSA values, % MC shed (66.7 ± 27.7) in voided urine and the fluorescent intensity (35.8 ± 5.7) were also the highest in GG5. Although the patient data points are too small to evaluate statistically, there appears to be a trend where the greater % composition of MC and the higher florescence aligned with increasing GG groups. This virtue, together with the noninvasiveness and the simplicity, makes this assay noteworthy and worthy of further investigation.

In total, 3 of the 62 patients with PCa and 1 of the 97 BPH subjects had Cu-64-TP3805 PET (Positron Emission Tomography) diagnostic imaging examination followed by either biopsy or radical prostatectomy as required by standard of care. Their voided urine was collected prior to any of these standard of care procedures. The results, given in Figure 3, depict that the urine assay was true positive for three PCa patients and true negative for the BPH patient as evidenced by the PET imaging, as well as with histopathologic examination of excised prostate tissue. The optical imaging data for 97 BPH subjects depicted that 87 were true negative for cells expressing VPAC. However, the remaining 10 samples were positive. Further evaluation of these ten subjects revealed that five of them had nephrolithiasis, three had renal cysts of various dimensions, one had chronic prostatitis and the other subject was on finasteride. Of the three patients with renal cysts, one had a renal transplant and another had squamous cell carcinoma of the bladder. Retrospectively, none of these ten subjects should have been included in this investigation. However, they were chosen because their PSA was <1.5 ng/mL. NKX3.1 immunohistochemistry examination of these ten subjects, excluding the one on finasteride, showed that the VPAC-positive cells present in their urine were not of prostate origin. The VPAC-positive cells were of prostate origin of the 10th subject on finasteride for the last 9 months. Although there was no follow-up on this subject, it may be possible that this subject may have clinically unrecognized PCa [16].

Statistical evaluation of urine assay data of the 97 BPH patients indicated that the specificity of the assay for BPH was 89.6% (95% CI, 81.9–94.9%), the positive predictive value (PPV) was 0.0% (95 CI, 0.0–30.9%) and the negative predictive value (NPV) was 100% (95%CI, 95.9–100%). The false positive rate due to other urologic conditions of the subjects was 10.3% (95% CI, 5.0–18.1%). Together, the data with this limited patient population strongly suggest that this simple, noninvasive, non-DRE voided urine assay can not only detect PCa with high accuracy and indicate the aggressiveness of the disease but also distinguish PCa from BPH.

## 4. Discussion

Globally, PCa presents a major health risk in men 60 years of age or older. It is the most heterogeneous disease in its clinical presentation, biochemistry, molecular biology and cell morphology. Further studies demonstrate that genomically and phenotypically distinct primary PCa can exist in a single patient. PCa detection is largely determined by the measurements of prostate-specific antigen (PSA) in the serum. Although sensitive, it is nonspecific and frequently leads to overdiagnosis and overtreatment. As a result, over the past few years, several urine-based assays have been developed, a few of which have been approved by FDA and by Clinical Laboratory Improvement Amendments (CLIAs). Recent meta-analyses of these assays, however, conclude that no single assay is perfect for a reliable and early diagnosis of PCa [17,18,19]. Not only do most require urine to be collected post DRE, massaging the prostate gland in a specific manner, but they also have low positive predictive values, are long to process and can be expensive. As a result, we are still far from a commonly accepted modality for early and accurate detection of PCa. Urine is an easily accessible biological fluid that contains a variety of substances, some of which are filtered from circulation. These include many metabolic waste products and small proteins secreted by various cell types. It also contains larger proteins and cells originating from the urinary tract downstream of glomerular filtration [20]. Low-speed centrifugation can separate the solid components of urine from the liquid fraction. The pellet usually contains cells, casts and debris, while the supernatant retains soluble components such as proteins, exosomes and cell-free nucleic acids that can be isolated and evaluated. All malignant lesions shed malignant cells [21]. The MCs shed from PCa are found in voided urine through prostatic pathways. MCs express on their surface genomic VPAC receptors in high density. Our assay is designed to target the VPAC receptors. The use of voided urine as a medium for clinical tests has several clear benefits. The process to collect a urine sample is noninvasive, painless and safe for patients, enabling frequent and repetitive large-scale collection for repeat testing if required. Urine contains exfoliated prostate epithelial cells and prostatic secretions because of the prostate’s anatomical relationship to the bladder and urethra. Urine usually contains several prostatic biomarkers, including both cell-associated and cell-free indicators. Given the potential for contamination from other body tissue types, urine as opposed to serum is better suited for prostate cancer use given the enriched cell population as the urine passes through the prostatic urethra [22]. While cytologic identification and descriptions of shed prostate cancer cells in urine samples have been reviewed and reported [23]; identification of prostate cancer cells based on morphology and immunohistochemistry alone is a difficult task even for very experienced cytopathologists given the scarcity of cells and overlap of cytologic findings with urothelial cell carcinoma [24]. While our previous work has established that VPAC is highly expressed on PCa cells, with 100% biomarker and histopathologic correlation with high sensitivity and specificity for prostate cancer [16], this work expands on those data to show differential expression based on the Gleason grade group at the time of radical prostatectomy, as well as to continue to demonstrate its ability to detect prostate cancer in urine. The original goal of this work was to develop a test with >95% confidence for a patient with elevated PSA to determine if this was due to PCa or benign prostate hyperplasia. Given the difference in fluorescence and % MC shed when stratified by the GG group, this test has the potential not only for diagnosis of a prostate cancer in a noninvasive manner, but also potential for prediction of aggressiveness of disease.

PCa and BPH can co-exist in 20% of men older than 50 years, confounding early and accurate diagnosis of PCa. Our data, although preliminary, suggest that this simple and truly noninvasive voided urine test has the ability to distinguish BPH from PCa and thereby help to minimize overdiagnosis and overtreatment. In our cohort of 97 BPH subjects, there were 10 false positive results. The subsequent revision of their urologic conditions revealed that five of the ten subjects had nephrolithiasis, three had renal cysts, one had chronic prostatitis and the other subject was on finasteride. Out of the three subjects with renal cysts, one had squamous cell carcinoma of the bladder and the other had a prior renal transplant. The exact mechanism by which some of the cells shed in voided urine of patients from such conditions may be expressing VPAC receptors is not clear at the time of this writing. The possibility, however, cannot be ruled out that the VPAC expression may be related to genomic transformation of the cells of renal origin, collected in the voided urine. Their NKX3.1 examination revealed that these cells were not of prostate origin. Our previous data confirm that inflammatory cells do not express VPAC [16]. Therefore, it is reasonable to rule that the inflammatory cells shed in voided urine of a patient with prostatitis may not have been a cause of the false positive results. The reason for false positive results of a patient treated with finasteride is also unclear. However, these cells were of prostate origin. There was no follow-up on this patient to rule out that he was free of PCa.

Our study is not without limitations, primarily it is a single institution study with low overall patient numbers, limiting a robust statistical analysis between Gleason grade groups. VPAC receptors are also known to be expressed in other solid tumors, such as breast, lung, central nervous system and neuroblastoma. While rare, it is possible that these results could be confounded by a yet unknown diagnosis of one of these tumors in the studied patient population. Despite these limitations, we believe that this optical imaging assay can be a simple, innovative, easy, affordable and specific way to detect VPAC receptors expressed by malignant PCa cells. We continue to collect patient samples to increase our sample size, and further improvements in the diagnostic utility of this test may be achieved when combined with other urinary-based prostate cell markers such as the STEAP family of proteins [25].

## 5. Conclusions

While our series is small, it builds on a body of evidence of the utility of alternative biomarkers to the commonly used PSA that may one day be able to help supplant the need for invasive biopsy for diagnosis, as well as potentially predict or provide prognostic indication for disease aggressiveness or to determine the presence of BPH.

## Figures and Tables

**Figure 1 diagnostics-15-00607-f001:**
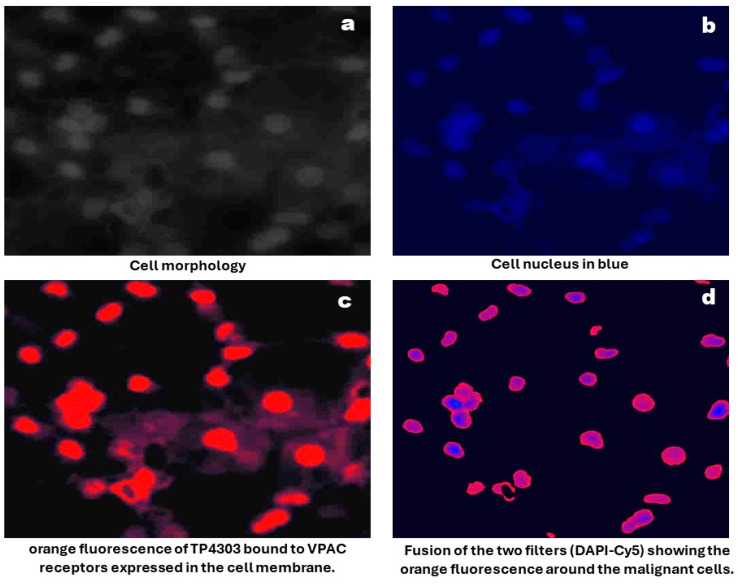
Optical imaging of cells prepared from voided urine.

**Figure 2 diagnostics-15-00607-f002:**
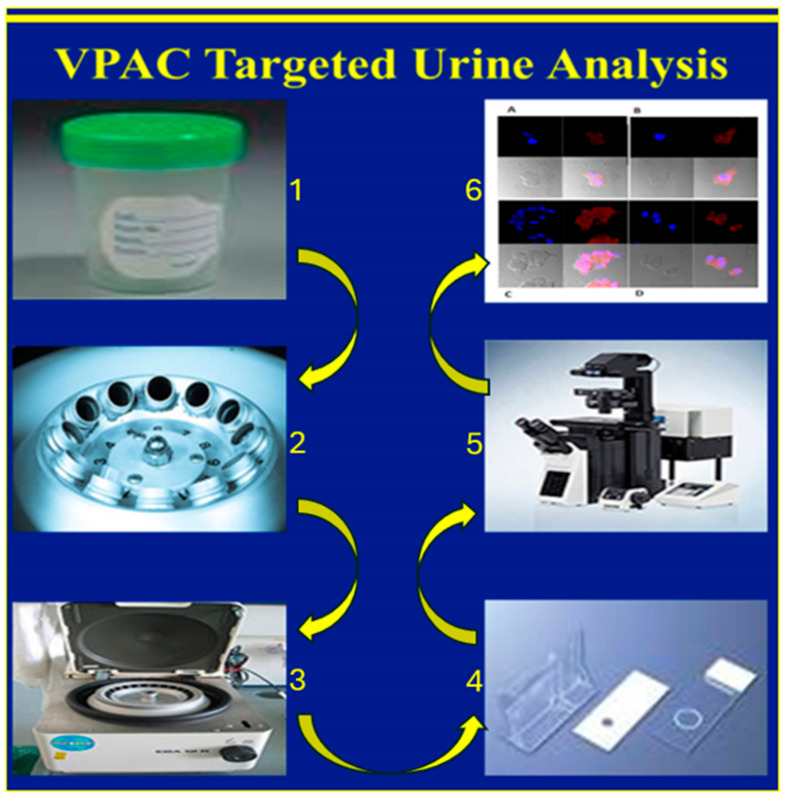
Urine is collected (1), concentrated (2) and cytocentrifuged, (3) cells on a glass slide are treated with TP4305 (4) and examined under the microscope (5), and malignant (with red fluorescence) and normal cells (blue) are identified (6).

**Figure 3 diagnostics-15-00607-f003:**
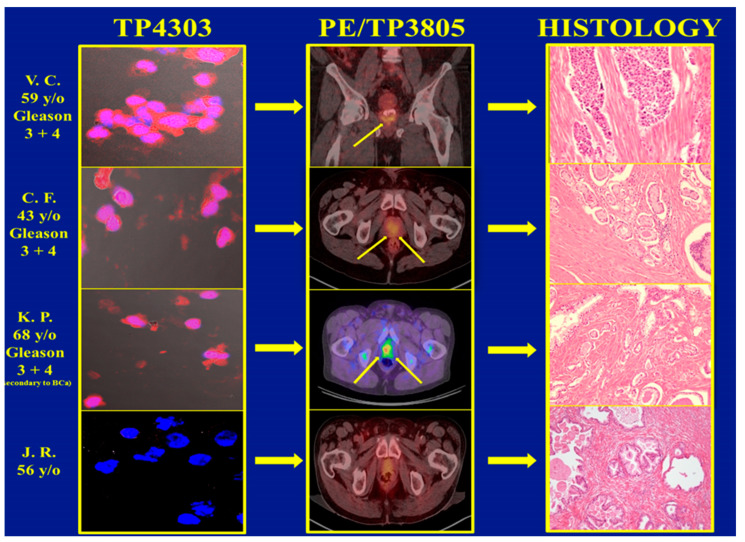
The top three images on the left are VPAC targeted optical images of malignant cells shed in the voided urine of three subjects with a Gleason score of 7 (3 + 4). The PET CT images in the middle panel depict PCa (arrows), and those on the right show their positive histopathology. The corresponding images of the fourth patient, clinically diagnosed with BPH, are normal.

**Table 1 diagnostics-15-00607-t001:** Patient clinical characteristics and results.

Patient Data and Results							
PCa Grade Group	No. of Patients	Age (yrs)	FSA (ng/mL)	Total Cells	Malignant Cells	% Malignant Cells	Fluorescence Intensity
		(Mean ± Standard Deviation)					
1	10	63.2 ± 5.9	6.52 ± 4.15	8588.3 ± 11679.9	2929.3 ± 3356.1	41.9 ± 28.1	31.8 ± 3.1
2	31	63.90 ± 7.3	7.21 ± 3.82	5562.2 ± 5207.2	2140.2 ± 2286.2	49.9 ± 29.6	33.7 ± 4.0
3	13	65 ± 9.2	13.18 ± 14.56	4892.3 ± 4143.2	2363.8 ± 2979.1	37.0 ± 27.2	33.0 ± 6.3
4	2	65.50 ± 6.3	6.55 ± 2.19	4685 ± 2616.3	1558.5 ± 1351.2	48.9 ± 56.1	32.2 ± 5.6
5	6	68.20 ± 4.1	50.20 ± 104.92	8961.3 ± 9877.6	7009 ± 8237.4	66.7 ± 27.7	35.7 ± 5.7

## Data Availability

The data presented in this study are available on request from the corresponding author.

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
