# Peer review of "Non-DRE Voided Urine Test to Diagnose Prostate Cancer: Updated Results"

_diagnostics, 2025, doi:10.3390/diagnostics15050607_

Round 1
Reviewer 1 Report
Comments and Suggestions for Authors
The topic of the manuscript is of high interest. PSA indicates various diseases and it is not specific to cancer. Therefore, it is actual to find approaches, which can diagnose PC among other diseases. Introduction is described clear and fully discuss the issue, but the next part of the study have some gaps, which should be filled before the publication. I have some recommendations, how to improve the article:
1. Please, add number of patients with histologically confirmed PCa and their age in «Study Population» section.
2. Considering the further validation of the approach, please add a reference on procedure optimization, which is described in “Sample Preparation and Imaging” section.
3. Statistical analysis can be applied to evaluate a relationship between PSA and GG (lines 130, 131). It would be more interesting to present correlation analysis (Spearman correlation) along with description of a relationship between PSA and GG.
4. Comorbidities can significantly influence the results. All comorbidities should be described in «Study Population» section for all patients, not only for 10: «Further evaluation of these ten subjects revealed that five of them had nephrolithiasis, three had renal sists of various dimensions, one had chronic prostatitis, and the other subject was on finasteride. Of the three patients with renal sists, one had renal transplant, and another had squamous cell carcinoma of the bladder». Patients with comorbidities is more interesting to study because it is closer to real clinical practice, but the information about comorbidities should be fully described. I do not think, that this sentence “Retrospectively, none of these ten 148 subjects should have been included in this investigation” is appropriate.
5. What does in means «Statistical evaluation” (line 160)? Which kind of statistical methods were used?
6. Authors can use Spearman correlation or Kruskal–Wallis test, which are appropriate for a small amount of data to conduct statistical analysis between Gleason grade groups (line 233).
7. Conclusion section (line 244): “Our study demonstrates a technique for sample collection and analysis that is simple 244 which could be easily replicated with little cost”. This conclusion is a result of previous work. Please, describe results of present work.
Reviewer 2 Report
Comments and Suggestions for Authors
Dear author
Thanks for your submission, I read your manusceipt and have some questions and recommendations
1- your manuscript should be reviewed by a native English speaker.
2- there are some spelling mistake in your manuscript like "sist"?! So revised your manuscript again
3-you did not mention anything about your control group.please mention the number of cells that shed in urine of your control group.
4- you choose men with normal psa as your control group, but as you know some patients had high psa without cancer. Do you have any evaluation for these patients?
5- you said that 5 patients with pbh and nephrolithiasis had MC in voided urine, what about patients in your case group, did you evaluate them for nephrolithiasis? In the other word, nephrolithiasis is a confounding factor for your study, how did you control this factor?
6- you should expand your exclusion criteria, for example exclude patients with genitourinary manipulations or infection, history of intercourse before urine sampling and ....
7- please mention sensitivity,specifity and positive and negative predictive value of your method for detection of prostate cancer
Round 2
Reviewer 1 Report
Comments and Suggestions for Authors
Authors answered all the question. As author opinion, statistical methods can not be applied due to unsufficient amoumt of data, probably, due to one of the group contains only 2 samples (3 samples in one group is minimal to apply non-parametric analysis). However, the group with unsufficial number of data can be excluded and the data can be analysed without the group, but the analysis will be not comprehensive. I hope, that authors will present further work with extended cohort o fparticipants. The present work can be accepted.